# Tailoring supercurrent confinement in graphene bilayer weak links

Rainer Kraft[1], Jens Mohrmann[1], Renjun Du[1], Pranauv Balaji Selvasundaram[1,2], Muhammad Irfan[3,4], Umut Nefta Kanilmaz[1,5], Fan Wu[1,6], Detlef Beckmann[1], Hilbert von Löhneysen[1,7,8], Ralph Krupke[1,2], Anton Akhmerov [3], Igor Gornyi[1,5,9] & Romain Danneau[1]

The Josephson effect is one of the most studied macroscopic quantum phenomena in condensed matter physics and has been an essential part of the quantum technologies development over the last decades. It is already used in many applications such as magnetometry, metrology, quantum computing, detectors or electronic refrigeration. However, developing devices in which the induced superconductivity can be monitored, both spatially and in its magnitude, remains a serious challenge. In this work, we have used local gates to control confinement, amplitude and density profile of the supercurrent induced in one-dimensional nanoscale constrictions, defined in bilayer graphene-hexagonal boron nitride van der Waals heterostructures. The combination of resistance gate maps, out-of-equilibrium transport, magnetic interferometry measurements, analytical and numerical modelling enables us to explore highly tunable superconducting weak links. Our study opens the path way to design more complex superconducting circuits based on this principle, such as electronic interferometers or transition-edge sensors.

[1] Institute of Nanotechnology, Karlsruhe Institute of Technology, D-76021 Karlsruhe, Germany. [2] Department of Materials and Earth Sciences, Technical University Darmstadt, Darmstadt 64287, Germany. [3] Kavli Institute of Nanoscience, Delft University of Technology, P.O. Box 4056, 2600 GA Delft, The Netherlands. [4] Department of Physics and Applied Mathematics, Pakistan Institute of Engineering and Applied Sciences, Nilore, Islamabad 45650, Pakistan. [5] Institute for Condensed Matter Theory, Karlsruhe Institute of Technology, D-76128 Karlsruhe, Germany. [6] College of Optoelectronic Science and Engineering, National University of Defense Technology, 410073 Changsha, China. [7] Institute of Physics, Karlsruhe Institute of Technology, D-76049 Karlsruhe, Germany. [8] Institute for Solid State Physics, Karlsruhe Institute of Technology, D-76021 Karlsruhe, Germany. [9] A.F. Ioe Physico-Technical Institute, 194021 St. Petersburg, Russia. Correspondence and requests for materials should be addressed to R.D. (email: romain.danneau@kit.edu)

S uperconductivity can be induced in a material by direct contact to a superconductor. This proximity effect allows the transmission of Andreev pairs from a superconducting electrode to another when these are close enough. The Josephson effect can then be measured as it is observed in tunnel junctions[1–3]. However, the tuning of the dissipationless current in such Josephson junctions is not possible without changing its geometry or temperature. By replacing the tunnel junction by a so-called weak link[4,5], i.e., any kind of conductive system, the supercurrent may flow over a much larger distance than the couple of nanometres of a tunnel barrier. The magnitude of the supercurrent mainly depends on the contact transparency, the disorder in the weak link and the temperature[4].

Many different types of materials and systems have been used as weak links, ranging from mesoscopic diffusive metallic wires[6], two-dimensional (2D) electron gas[7], graphene[8], topological insulators[9–14] and quantum dots[15], as well as atomic contacts[16]. When graphene is utilised as a weak link, the Josephson effect can be tuned by electrostatic gating[8,17–23], and thanks to edge connection which provides very low contact resistance[24], it is possible to measure large supercurrent amplitudes as well as ballistic interferences[25–29]. However, in spite of these excellent predispositions to mediate superconductivity, a full control of the supercurrent both in its amplitude and spatial distribution has not been demonstrated up to now. One of the reasons behind this is the difficulty to confine charge carriers in graphene due to the absence of back scattering and Klein tunnelling[30]. The use of bilayer graphene (BLG) could circumvent these problems since it is possible to engineer an electronic band gap by breaking the lattice inversion symmetry of the AB-stacked bilayer[31,32]. Indeed, by means of local gating, BLG can provide a way to shape the supercurrent distribution and allow a complete monitoring of proximity induced superconductivity. Here, we have used edge connected BLG-hexagonal boron nitride (hBN) heterostructures as a medium for induced superconductivity, and use a quantum point contact (QPC)-like geometry to study supercurrent confinement.

## Results

### Reading a dual gate map and inducing a 1D constriction.

The sample geometry used in this study is depicted in Fig. 1. Following the fabrication method of Wang et al.[24], we employ BLG encapsulated between hBN multilayers connected from the edge of the mesa with superconducting titanium/aluminium electrodes. The constriction is realised by inducing displacement fields between an overall pre-patterned back-gate (BG) and a local top-gate designed in a QPC-like split-gate (SG) geometry (see Fig. 1). Two devices were measured which show similar behaviour, here we present the data based on the shortest sample (details on the sample fabrication are presented in Supplementary Note 1).

The normal state characteristics of our sample show a residual charge carrier density as low as $2.6 \times 10^{10}\ \mathrm{cm^{-2}}$, well-developed Landau fans in magnetotransport experiments, as well as multiple Fabry–Pérot interferences generated by the charge carriers travelling back and forth within the several cavities formed in our system (see Supplementary Notes 2 and 3, Supplementary Figs. 1–3 for full analysis). Figure 2a, b display resistance maps as a function of SG and BG voltage measured in the normal and superconducting state respectively (i.e., at 20 mT and zero magnetic field). In both cases, distinct deviations from the expected quadrants formed in lateral npn-junctions corresponding to the differently doped regions[33–35] are clearly visible (unipolar and bipolar regions NNN, PPP and NPN, PNP respectively).

In BLG dual-gated devices, the displacement field is used to break the lattice inversion symmetry of the AB-stacked bilayer: the two layers being at different potentials a band gap opens[31,32], inducing an insulating state with strongly suppressed conductivity. The resistance then raises monotonically with increasing displacement field as the band gap develops[33–35]. Here, we observe a non-monotonic change of the resistance which first increases and then drops after reaching a maximum while following the displacement field line (i.e., when the displacement field generated by the BG and SG, respectively $D_b$ and $D_t$ are equal, at $\delta D = D_b - D_t = 0$[36]). In addition, the resistance peak does not follow the displacement field line which is indicated by the grey arrow as depicted in Fig. 2a, b, but diverges into the bipolar regions (NPN and PNP). This trend is already noticeable in the normal state resistance (Fig. 2a), but becomes strikingly evident in the superconducting state (Fig. 2b). This unexpected behaviour can be understood as the competitive action of BG and SG within the constriction. As the displacement field increases, the charge carrier density mostly driven by the BG becomes less and less affected by the stray fields developed by the SG, which cannot compensate the influence of the BG on the channel region. Instead of being maximum along the displacement field line[33–35] (marked as a diagonal arrowed line on the gate maps), the resistance increases up to a maximum then decreases as plotted in Fig. 2c for both normal and superconducting states. Consequently, the device remains highly conductive in contrast to the pinch-off characteristic of gapped BLG with full-width top-gate. Instead, the maximum resistance deviates from the displacement field line and 'bends'. The bent line of the resistance peak results then from the required overcompensation of the SG voltage to diminish the induced charge carriers within the channel region. In the superconducting state, the resistance follows the same tend to finally drop to zero (Fig. 2c).

However, this imbalance between applied SG and BG voltages starts to induce charge carriers of opposite sign in the dual-gated cavities, resulting in pn-junctions. As a consequence, the bipolar regions become then subdivided into two parts depending on the

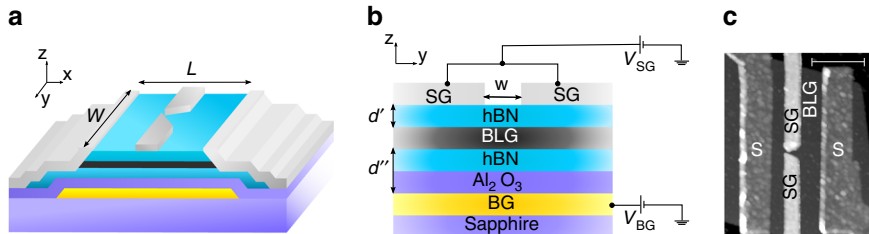

**Fig. 1** Device geometry. **a** 3D Schematics of the device and **b** cross-sectional view as a cut through the dual-gated region. The device consists of a hBN-BLG-hBN heterostructure (with a bottom and top hBN multilayer of ~35 and ~38 nm thick) on a pre-patterned overall back-gate (BG) covered with a 20 nm thick $Al_2O_3$ and a split-gate (SG) on top of the heterostructure. The superconducting leads are edge connected to the mesa. The width $W = 3.2\ \mu m$ and length $L = 950$ nm while the distance between the two fingers of the split-gate $w \sim 65$ nm (and $d' \sim 38$ nm and $d'' \sim 55$ nm). **c** AFM image of the device. Scale bar is 1 μm

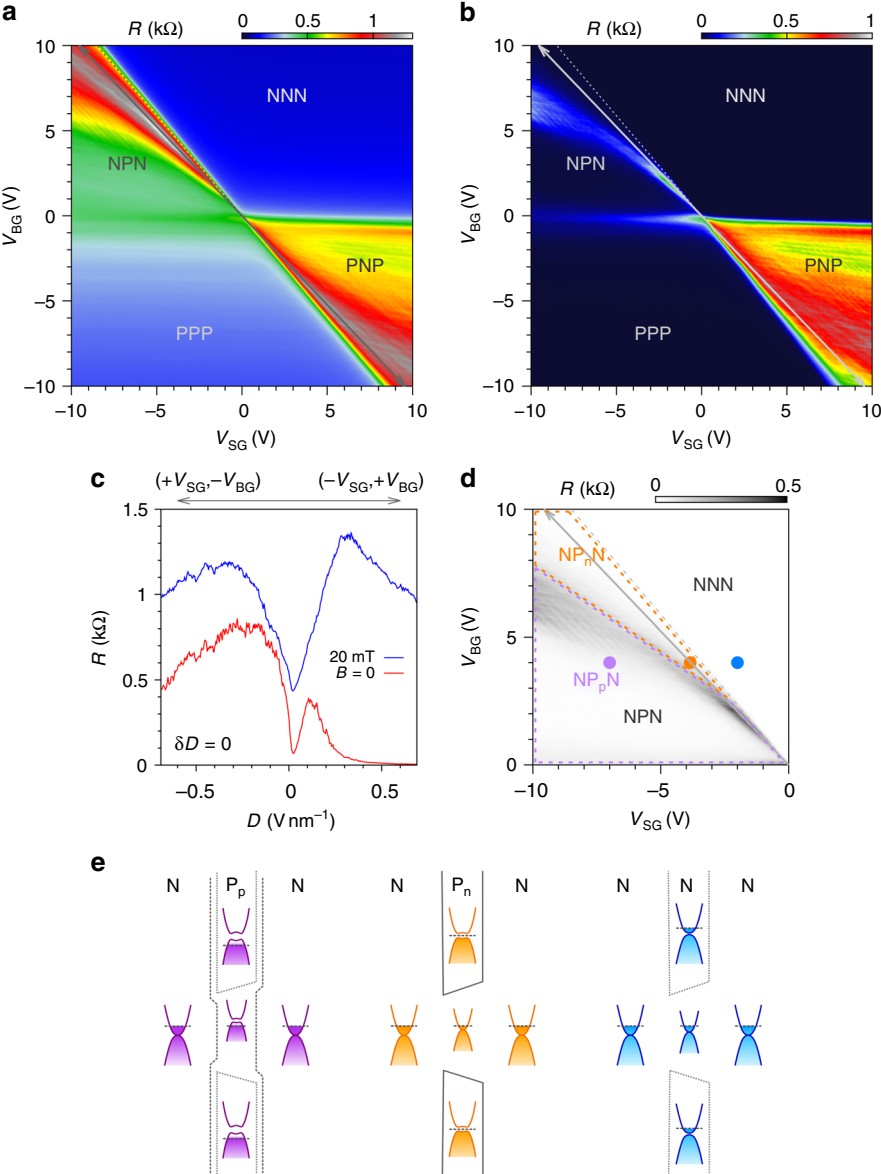

**Fig. 2** Formation of the constriction: resistance gate map analysis. **a** Resistance map as a function of split-gate and back-gate voltage, $V_{SG}$ and $V_{BG}$ respectively, measured at ~25 mK in the normal state ($B = 20$ mT) with a maximum current of 2.6 nA. The arrow marks the displacement field line along which the charge carrier density in the dual-gated region is zero. The dashed line indicates the transition when $E_F$ is tuned from the conduction band into the induced band gap, highlighting the crossover to a confined system. **b** Resistance map versus $V_{SG}$ and $V_{BG}$ measured at ~25 mK in the superconducting state ($B = 0$) with a maximum current of 2.5 nA. **c** Normal (blue curve) and superconducting (red curve) state resistance measured along the displacement field line. **d** Zoom-in on the upper left part of the resistance map in the superconducting state (**b**), where the different regime areas are enlightened. **e** Schematics of the spatially resolved energy band diagrams of our QPC geometry where top-views of the device refer to the three different regimes of panel **d**

doping in the constriction (denoted by a sub-label like NP$_n$N, see Fig. 2d). The QPC-like structure can then be driven in an 'open' (the 1D channel doping is of the same type as the 2D reservoirs) or 'closed' (the 1D channel doping is of opposite type as the reservoirs forming a non-uniform potential barrier) regime.

The schematics in Fig. 2e summarise the different scenarios which govern the behaviour of such an electrostatically induced constriction, i.e., the formed 1D constriction area NP$_n$N, the unipolar regime NNN and the non-uniform NP$_p$N junction. It is important to note that the overall resistance remains higher on the p-side (PPP and PNP) due to the slight n-doping provided by the leads which create a pn-junction at each contact. This

becomes particularly clear in the superconducting state where the PNP region remains resistive, while a large part of the NPN section displays a zero resistance state. For this reason, we focus on the NPN area and in particular on the NP$_n$N part where we can study the supercurrent flowing through the constriction.

**Supercurrent analysis**. Now we describe how to control both supercurrent amplitude and spatial distribution using our SG geometry. We have seen in the previous section that our device becomes superconducting in the area where the constriction is formed, namely the NP$_n$N region. One way to verify our

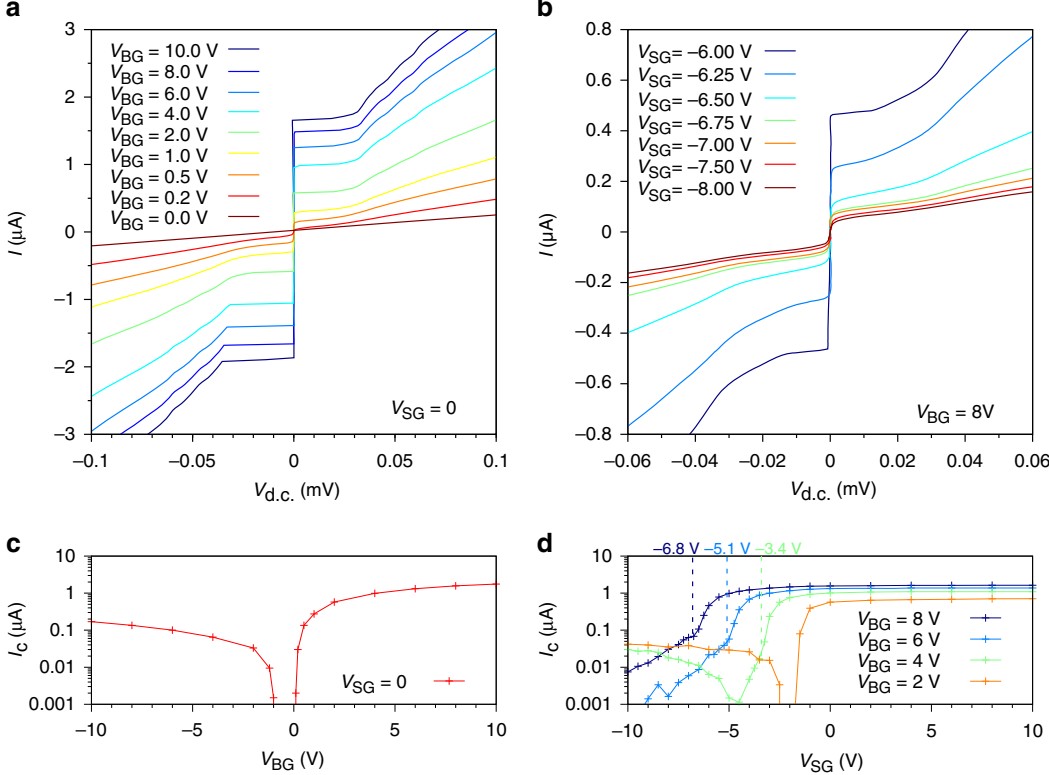

**Fig. 3** Gate-controlled current in a superconducting BLG weak link. **a** $I$-$V$ curves for different $V_{BG}$, i.e., densities, characterising the 2D system at $V_{SG} = 0$. **b** $I$-$V$ curves at fixed back-gate voltage $V_{BG} = 8$ V for various displacement fields $D$ in the dual-gated region, i.e., for split-gate voltages close to the transition from NNN to NP$_n$N. **c** Back-gate voltage dependence $V_{BG}$ of the critical current $I_c$. **d** $I_c(V_{SG})$ for constant charge carrier densities (i.e., constant $V_{BG}$)

hypothesis consists of probing the critical current $I_c$ which corresponds to the maximum supercurrent that a weak link can support before switching to a resistive state (see method section for a description of the critical current extraction procedure). $I_c$ being extremely sensitive to any external perturbations such as magnetic field, potential landscape inhomogeneities or thermal excitation, drastic changes of the confinement should be clearly observed. Indeed, the variation of the normal state resistance is directly reflected in the supercurrent amplitude. For example, small oscillations in the resistance produced by Fabry–Pérot interferences are directly detected in the supercurrent[25,26,29,37] (see Supplementary Fig. 3). Here, we focus our attention on the effect of the 1D constriction on the supercurrent amplitude.

The amplitude of the supercurrent can be monitored by tuning the charge carrier density with the overall BG voltage $V_{BG}$. In Fig. 3a the current-voltage characteristics are shown in the absence of a constriction, i.e., for a uniform 2D weak link at $V_{SG} = 0$. The supercurrent evolves from zero at the charge neutrality point up to a measured maximum of 1.86 μA at high charge carrier density $n \approx 4 \times 10^{12}$ cm$^{-2}$ (i.e., $V_{BG} = 10$ V). It is important to note that the $I$-$V$ characteristics only display a rather limited hysteretic behaviour (see Supplementary Fig. 4) visible only at large charge carrier density corresponding to a weakly underdamped junction within the resistively and capacitively shunted junction model[3]. When the Fermi level lies in the valence band ($V_{BG} < 0$), the weak link is disturbed by the presence of the pn-junctions, which strongly suppresses the supercurrent by an order of magnitude (~200 nA at $V_{BG} = -10$ V). This is clearly seen in Fig. 3c where the critical current $I_c$ is plotted as a function of the BG voltage $V_{BG}$ (supercurrent amplitude values are summarised in Supplementary Table 1; see also Supplementary Fig. 4 for $I$-$V$s at negative BG voltage $V_{BG}$).

Figure 3b displays a series of $I$-$V$ curves at fixed charge carrier density (here at $V_{BG} = 8$ V) for different SG values in the vicinity of the NP$_n$N area. When approaching the formation of the constriction, $I_c$ decreases rapidly until $V_{SG} \sim -6.65$ V. At this point, the Fermi level underneath the SG is positioned in the gap. Therefore, charge carriers can only flow through the 1D constriction. Beyond the formation of the constriction, $I_c$ decreases in a much slower fashion. The extracted critical current $I_c$ is plotted in Fig. 3d as a function of the SG voltage $V_{SG}$ at different densities. At small densities, i.e., $V_{BG} = 2$ V (orange curve in Fig. 3d), the starting point of the NPN region appears early in gate voltage and the supercurrent is switched off. Then, the Fermi level in the constriction which remains mainly driven by the stray fields of the SG moves toward the valence band. Due to the close proximity of the SG, the stray fields are strong enough to close the channel. A small supercurrent can be detected despite the presence of a weak pn-junction as depicted in Fig. 2d (NP$_p$N area). In contrast, at higher densities the BG starts to electrostatically dominate the constriction region. The creation of the 1D channel is directly reflected in the sudden change of slope of $I_c(V_{SG})$ curves (blue and dark blue curves in Fig. 3d, the change of slope being marked by dotted lines). The supercurrent through the channel is then only slowly reduced with increasing SG voltage owing to the narrowing of the channel by the stray fields. Once the channel is created, the amplitude of the supercurrent drops way below 100 nA, while multiple Andreev reflections completely vanish (see Supplementary Note 4 and Supplementary Fig. 4). At intermediate density (green curve in Fig. 3d), the channel is first created (rapid drop in $I_c(V_{SG})$ then change of slope marked by the dotted curve), then closed with the Fermi level positioned in the gap (supercurrent switched off), to finally form a non-uniform pn-junction as depicted in Fig. 2e

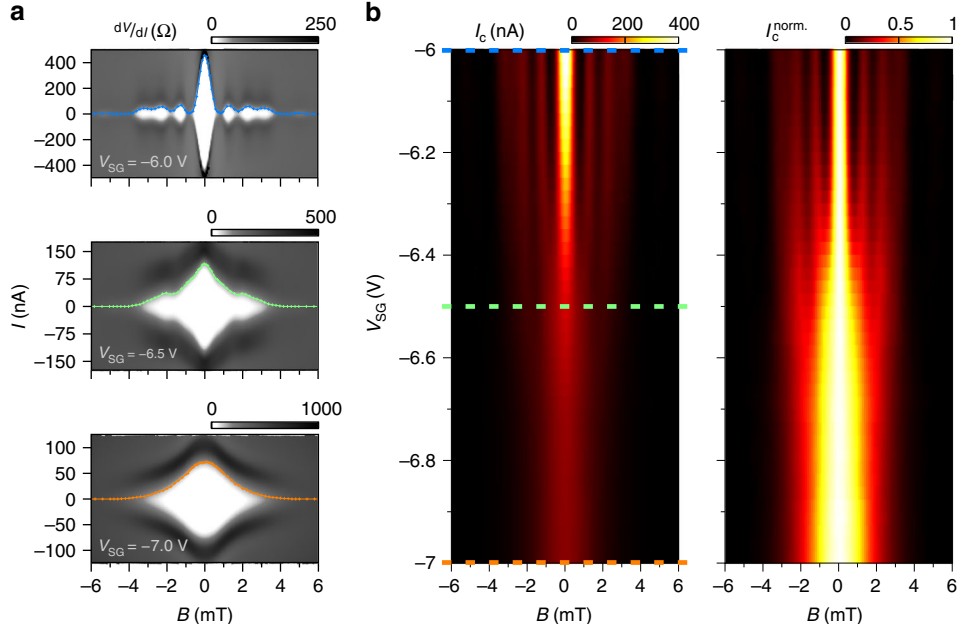

**Fig. 4** Magnetic interferometry study of the transition from 2D to 1D confinement of the supercurrent. **a** Grey-scale map of the differential resistance d$V$/d$I$ versus magnetic field $B$ and current $I$. The coloured dotted lines correspond to the extracted $I_c$. These measurements are taken at three different split-gate voltage values ($V_{SG} = -6$, $-6.5$ and $-7$ V) at constant charge carrier density ($V_{BG} = 8$ V). Drastic change in the interference pattern is observed highlighting a clear transition from 2D to 1D confined supercurrent. **b** Critical current amplitude $I_c$ (left panel) and normalised critical current amplitude $I_c^{norm.}$ (right panel) mapped as a function of magnetic field $B$ and split-gate voltage $V_{SG}$. The transition from a beating pattern (Fraunhofer-like) to a monotonically decaying pattern is visible confirming the continuous change in the supercurrent confinement from 2D to 1D. The coloured dashed lines correspond to the split-gate values where the d$V$/d$I$($V_{SG}$,$B$) maps were measured in panels **a**

(NP$_p$N area). We note that, on the presented device, we observe features visible in the normal and the superconducting state which could be related to quantised conductance and supercurrent (see Supplementary Note 5, Supplementary Figs. 5 and 6) as predicted for ballistic supercurrents in QPCs[38–40].

**Magneto-interferometry.** The supercurrent density distribution across the sample width can be explored by probing its interference pattern[41], in response to a perpendicular magnetic flux penetrating the junction[5,13,27,42–48]. Therefore, by changing the geometry of the system one can observe a large variety of interference patterns directly related to the supercurrent density distribution[5]. As recently shown[13,27], superconducting interferometry is a powerful tool to probe confinement where the current density distribution can be extracted by complex Fourier transform following the approach of Dynes and Fulton[42]. However, this technique of recovering the supercurrent assumes that it is carried strictly in a direction normal to the superconducting electrodes, and therefore does not apply to our device because of the non-uniform supercurrent density in $x$-direction and its small aspect ratio in the QPC regime.

Here, we show that the magnetic interference pattern indicates clear signatures of the supercurrent confinement. Figure 4a exhibits a series of resistance maps versus current and magnetic field at constant density ($V_{BG} = 8$ V). By increasing $V_{SG}$, i.e., the confinement, a progressive change of the interference pattern is observed as the SG is tuned and the 1D constriction forms. First, a beating pattern appears, resembling Fraunhofer-like interference (upper panel) when the system remains two-dimensional. Then the interference pattern turns to a 'lifting lobes' shape just before the formation of the constriction (middle panel). Finally a non-beating 'bell-shaped' pattern is formed while the supercurrent flows only through the confined 1D constriction (lower panel). We note that the transition from a beating to a non-

beating pattern occurs on a rather narrow voltage range $-7$ V $< V_{SG} < -6$ V (at $V_{BG} = 8$ V, additional data at $V_{BG} = 4$ V are shown in Supplementary Note 6 and Supplementary Fig. 7). In Fig. 4b we can observe a map of the critical current $I_c$ (left panel), as well as the critical current normalised to the maximum critical current (at $B = 0$) $I_c^{norm.} = I_c/I_c(0)$ (right panel) as a function of magnetic field $B$ and SG voltage $V_{SG}$, allowing a more accurate vision of the transition from 2D (beating pattern) to 1D ('bell-shaped' pattern). Each horizontal slice of such maps corresponds to the extracted critical current (or normalised critical current) of a single magnetic interference pattern (additional remarks and data can be seen in Supplementary Note 6 and Supplementary Fig. 7). We note that such non-beating pattern has been observed in rectangular superconducting weak links with low aspect ratio[46–48].

In order to gain deeper understanding how the magnetic interferences should evolve with the creation of a 1D constriction into a 2D system, we have designed an analytical model where we calculate the Josephson current through the sample in the presence of a magnetic field $B$ (see Supplementary Note 7, Supplementary Figs. 8 and 9 for details), using a quasi-classical approach (as in ref. [44,49,50]) with an additional input given by the presence of a QPC-like structure in the middle of the device (see the geometry used in Fig. 5a). We have used our analytic expression to fit the maximum critical current as a function of magnetic field (see Fig. 5b). The theoretical critical current (magenta curve) is matched to the experimental data $I_c(B)$ (turquoise crosses) by scaling the curve by a factor of the extracted maximum critical current $I_c(0) = 43.5$ nA using a junction area of $\sim 6.08 \times 10^{-12}$ m$^2$ with a total junction length of $\tilde{L} = L + 2\lambda_L = 1.90$ μm, where $\lambda_L$ is the London penetration depth ($\lambda_L \sim 450$ nm). Our model follows clearly the experimental data $I_c(B)$ which, once again, proves that the supercurrent has been strongly confined in our QPC edge connected BLG. We also

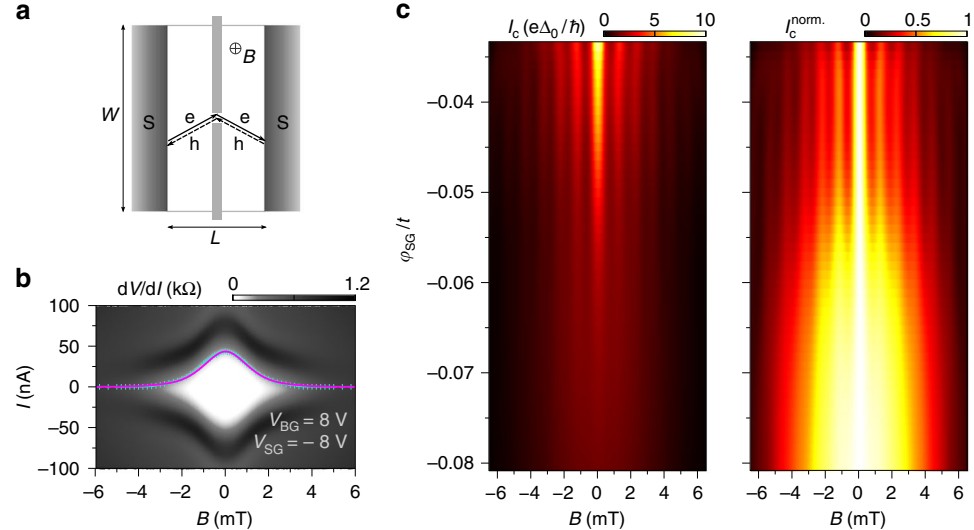

**Fig. 5** Modelling supercurrent confinement. **a** Schematic of the superconducting weak link with a quantum point contact like geometry used for our analytical model. **b** Differential resistance d$V$/d$I$ versus magnetic field $B$ and current $I$ including the extracted critical current $I_c$ (turquoise crosses) fitted with our analytical model (magenta line) when the 1D constriction is formed (at $V_{BG} = 8$ V and $V_{SG} = -8$ V). **c** Numerical simulations of critical current amplitude $I_c$ (left panel) and normalised critical current amplitude $I_c^{norm.}$ (right panel) mapped as a function of magnetic field $B$ and split-gate strength $\varphi_{SG}$ showing the transition from 2D to 1D of the magnetic interferences. The x-axis is rescaled to magnetic field $B$ using the parameters extracted by fitting the numerical simulation to the experimental data at $V_{SG} = 0$ (see Supplementary Note 7, Supplementary Figs. 8 and 9, Supplementary Note 8, Supplementary Figs. 10 and 11 for details)

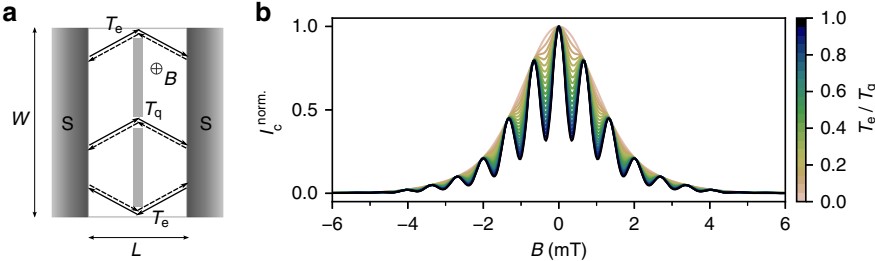

**Fig. 6** Effect of the edge currents on the interferometric pattern. **a** Schematic of the superconducting weak link with a quantum point contact like geometry including possibility of edge currents used for our analytical model. **b** Calculated $I_c^{norm.}$ versus magnetic field $B$ for different values of the transmission coefficient ratio $\mathcal{T}_e/\mathcal{T}_q$

performed tight-binding simulations using Kwant package[51] of $I_c$ as a function of magnetic field $B$ and SG strength $\varphi_{SG}$ (in units of the intralayer hopping constant $t$) in Fig. 5c (see Supplementary Note 8, Supplementary Figs. 10 and 11 for details), which are in good qualitative agreement with our experimental data of Fig. 4b.

Finally, from the magneto-interferometry experiments, no obvious signs of induced current through topological channels appearing due to AB stacking faults[52] or edge states[27,53,54] have been detected. In order to verify this assumption, we have estimated what would be the effect of additional current flowing at the edges (with a transmission $\mathcal{T}_e$) while the current is confined by the QPC (with a transmission $\mathcal{T}_q$) as depicted in Fig. 6a. Our analytical model (see Supplementary Note 9 and Supplementary Fig. 12 for details) shows that the presence of edge states develops a beating on top of the pattern formed by the 1D confinement. When we look at the expected magneto-interferometric pattern as displayed in Fig. 6b, we clearly see the effect of the additional current which modulates periodically with magnetic field $B$ the amplitude of $I_c^{norm.}$. The modulation becomes visible for $\mathcal{T}_e \sim \mathcal{T}_q/100$. In absence of edge states, $\mathcal{T}_e = 0$, we recover our previous result shown in Fig. 5b. Thus, it is clear that if one would be confronted to the presence of current flowing through the edges (or any other current path) in our QPC geometry, the magneto-interferometric pattern should be directly affected for, at least, current values down to a hundredth of the current through the constriction. Here, we have measured a confined supercurrent down to ~20 nA. Therefore, one should be able to detect the presence of edge currents down to 200 pA or less, (i.e., 25 times less than measured in ref. [53]).

At BG and SG values of $V_{BG} = 8$ V and $V_{SG} = -7.6$ V respectively, the induced displacement field of $D \approx 0.56$ V/nm (one order of magnitude larger than in ref. [53]) opens a significantly large band gap $E_g \approx 85.4$ meV (for the gap extraction see Supplementary Note 2) and no edge current is detected by mangeto-interferometry. The absence of edge contribution to the total current could be explained by the edge-state localisation in the case of large band gap as reported in ref. [55]. The characteristic value of the localisation length was found to be about tens of nanometres, which is much smaller than the top-gated region in our experiment, thus implying strong suppression of edge currents.

On the basis of our model, the presence of conducting edge channels should be detectable within our experimental conditions. However, the relatively large band gap may cause the suppression of the edge currents which could be the reason why we do not observe their presence in our measurements.

## Discussion

In this work, we have demonstrated a full monitoring, both spatially and in amplitude, of the supercurrent in a clean and edge connected hBN-BLG-hBN heterostructure. In a SG geometry we have explored the consequences of the 1D confinement on the supercurrent and on its magnetic interferences. Thanks to in turn, the possibility to locally engineer an electronic band gap in BLG, the injection of a large and fully controllable critical current, the ultra-low disorder of encapsulated hBN-BLG-hBN heterostructures and the absence of edge currents, we have designed a unique platform allowing the creation of new types of superconducting circuits based on fully tunable weak links which can be controlled by the combination of top-gates and BGs.

## Method

**Experimental**. The low-temperature electrical measurements were performed in a Bluefors LD250 $^3$He/$^4$He dilution fridge at a base temperature of ~25 mK. All dc-lines were strongly filtered using 3-stage RC-filters with a cut-off frequency of 1 kHz, as well as PCB-powder filters with a cut-off frequency of about 1 GHz. The differential resistance/conductance data was measured using standard low-frequency (~13 Hz) and various low excitation (between 1 and 10 μV), the gating and the out-of-equilibrium measurements were performed using ultra-low noise dc-power supply from Itest. The normal state was obtained by applying a perpendicular magnetic field of 20 mT. The experiments were performed within several thermal cycles (room temperature ⇌ milli-Kelvin temperature). Data have been reproduced and implemented in each cooldown.

**Data treatment and $I_c$ extraction**. The critical current $I_c$ is extracted using a voltage threshold method, where the threshold is set to 1 μV. The two adjacent data points of recorded $I$-$V$s right before and after the threshold are evaluated, and $I_c$ is determined by linear extrapolation in the current of these two points depending on the difference of the voltage drop with respect to the threshold. The extracted critical current is corrected by subtracting the artificial offset that is produced by this method.

**Data availability**. The data that support the findings of this study are available from the corresponding author upon request.

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

## Acknowledgements

The authors thank A. Mirlin, M. Titov and W. Wernsdorfer for fruitful discussions. This work was partly supported by Helmholtz society through program STN and the DFG via the projects DA 1280/3-1 and GO 1405/3-1. A.A. and M.I. acknowledge support of the European Research Council, and the Netherlands Organisation for Scientific Research (NWO/OCW), as part of the Frontiers of Nanoscience program. We also acknowledge the support from the FLAG-ERA JTC2017 Project GRANSPORT.

## Author contributions

R. Kraft performed the experiments with the support of J.M., R.Du, P.B.S., F.W. and R. Danneau R. Kraft fabricated the devices with the support of J.M. U.N.K. and I.G. designed the analytical model. M.I. and A.A. performed the numerical calculations. All authors discussed about the results. R.Danneau and R. Kraft performed the data analysis and wrote the paper. R. Danneau designed and planned the experiments.

## Additional information

**Competing interests:** The authors declare no competing interests.

