## [Peer Review File · Nature Communications]

Reviewers' comments:

Reviewer #1 (Remarks to the Author):

The authors investigated the supercurrent in bilayer graphene weak links using a QPC-like geometry. They used the split-top gates to create one-dimension constrictions. The ability to control the dimension of the weak links from two to one dimension allows them to tune critical currents as well as density profile of the supercurrent. The authors also studied the interference pattern of the supercurrent vs magnetic field as the weak link dimension changes from two to one dimension by varying split gate voltage and back gate voltage. The results agree well with their theoretical model.

The data are of high quality and the analysis on magneto-interferometry is solid. I have the following questions/concerns which the authors should address before the paper can be published:

1) The ability to create 1D constriction in this experiment relies on an insulating state in BLG which can be induced by displacement field. However, such insulating state is very hard to achieve in BLG due to small band gap generated by displacement field and potential fluctuations from disorders. The authors should estimate the size of the band gap from displacement field (see Nature 459, 820-823 (2009)) and the potential fluctuation from a residue charge inhomogeneity and see if the band gap is larger than the potential fluctuation or not.

2) From Fig. 3a, b in the supplementary information, Fabry-Perot oscillations exists in NP_nN regime and the period of these oscillations corresponds to the cavity underneath the split gates. Doesn't this mean that BLG under the split gates in the NP_nN regime is not completely insulating since electrons can still propagate through it. Therefore, the device does not exhibit 1D constriction in the NP_nN regime.

3) The authors attribute the step-wise decrease of the critical current to ballistic supercurrents through QPC (Fig 3e in main text). However, they do not observe any signature of 1D subbands in the normal state. Do you have any explanation why 1D subbands would be more pronounced in the superconducting state? Can the "step-wise decrease" be the consequence of Fabry-Perot oscillations (It just happened to look like steps when in fact it's due to oscillations of resistance in the normal state). Figure 3d in the supplementary information also shows oscillations in critical current with similar periods as the one in Fig 3e of the main text.

4) The authors mention Fig 4c at the last line of page 6 of the main text but there is no Fig 4c.

Reviewer #2 (Remarks to the Author):

The manuscript by Kraft et al. reports a systematic and detailed investigation of the spatial and amplitude variation of the superconducting critical current (I_c) in hBN-encapsulated bilayer graphene obtained via the application of back- and top-gate voltages. The results obtained are technically correct and the analysis of the measurement data is indeed very detailed.

However, there have been many studies recently which have explored the possibility of controlling the superconducting I_c in graphene-based Josephson devices by tuning the charge carrier type and density through the application of gate voltages. Other research groups have demonstrated changes in the graphene normal-state conductance from a few-mode to a single-mode channel (e.g. P. Rickhaus et al., Nano Lett. 15 (9), 5819–5825 (2015)), as well as investigated the spatial variation of I_c with bias voltage in superconducting S/N/S junctions of different lengths where hBN-encapsulated graphene is the weak link (e.g. I.V. Borzenets et al., Phys. Rev. Lett. 117, 237002 (2016) or M.B. Shalom et al., Nat. Phys. 12, 318-322 (2016)).

From this point of view, the manuscript by R. Kraft et al. seems a natural extension of similar studies already reported in the literature for single-layer graphene to the case of bilayer graphene. Although it can be argued that this work is different compared to other studies on the modulation

in I_c as a function of carrier density and type in bilayer graphene (e.g. M.T. Allen et al., Nat. Phys. 12, 128-133 (2016)) and that a systematic study of the spatial variation of I_c is addressed, I think that it does not meet the elements of novelty and originality necessary for publication in Nature Communications and therefore I recommend its publication in a more specialist journal.

Lastly, I think the authors may find useful to address the following minor points which would make the text of the manuscript clearer to the reader:

1. In paragraph on page 2 starting with "As a consequence, the bipolar region.." the text would be clearer if the authors explained all the labels used (NPnN, NPPN etc.) – which are actually clearly defined only in the caption of Figure 2.
2. The scale bar in Figure 2c does not look correct, as it seems that Figure 2c should contain some areas coloured in green in Figure 2b – which should correspond to a resistance value of 0.3-0.4 k Ω according to the scale bar in Figure 2b.
3. In the caption of Figure 3e, I think the authors mean 'differential resistance' other than just 'resistance'.
4. In the paragraph starting with "First, a beating pattern appears.." on page 5 the authors should clarify that the evolution from a Fraunhofer-like pattern into a bell-shaped I_c Vs. H curve happens for increasing gate voltages, as this only becomes clear to the reader after looking the captions of Figure 4a.
5. In the caption of Figure 4, the split-gate voltage values should be negative, in order to be consistent with what reported in Figures 4a and 4b as well as in Figure 1c.
6. At the end of page 6, the authors explain how $I_{c, \text{norm}}$ is calculated, but they never define $I_{c, \text{norm}}$ explicitly. This should be done, as $I_{c, \text{norm}}$ is used for the first time in the caption of Figure 5 but without any definition.
7. In the last section of the supplementary information, the authors state that the temperature used in the numerical simulation is higher than the actual measurement temperature, as the relevant scale is the thermal length. Using a lower temperature in the numerical code, however, should have an effect at least on the amplitude of the calculated I_c and perhaps on its spatial dependence as well. It would help if the authors could further clarify on this point and perhaps add to the SI file another $I_{c, \text{norm}}$ versus applied field contour plot, similar to that shown in Figure 5c, but calculated at a different temperature (e.g. $k_B T = \Delta_0/10$).

Reviewer #3 (Remarks to the Author):

The authors study encapsulated bilayer graphene contacted by superconducting titanium/aluminium electrodes. They fabricate a split gate on top of the sample (Figure 1), which eventually allows them to study superconducting current through a 1D constriction. They measure the normal state resistance and the critical current vs the back gate and the split gate in Figure 2. Different regimes are identified, depending on whether the regions of the channel and that under the split gate are P or N doped. In particular, the maps of Figure 2 allow them to find the range of parameters in which the region under the gate is gapped and depleted, but the channel is doped and conductive. This situation occurs when the back gate voltage is high enough (beyond about 6 V), and the split gate voltage is negative enough to shut off the conductance bypassing the channel formed by the split gate. In this regime, the residual critical current of the order of 10-100 nA is observed (Figure 3d,e).

At this point the author make a rather ambitious claim that the critical current is quantized: "Importantly, despite the absence of signs of 1D subband formation while shrinking the constriction in the normal state, the critical current decreases in a step-wise fashion (see Fig. 3e) as predicted for ballistic supercurrents in quantum point contacts [38–40]." I totally agree with them that without observing the quantized conductance, it is strange to expect the critical current quantization. To substantiate their claim, I believe that it is crucial to demonstrate this behavior in more than one sample. Also, it would be important to show explicitly how the map of Figure 3e is

converted to a graph of critical current vs gate voltage. Finally, the normal state resistance in the range shown in Figure 3e appears to be about 500 Ohm, corresponding to conductance of about $50 e^2/h$. It seems inconceivable that the critical current would be quantized and show only a few first few quantized steps as it may be suggested by the shape highlighted in Figure 3e.

The rest of the paper (Figure 4) and most of the supplementary information describe the magnetic interference pattern. The similarity between the data and numerical simulations appears very conclusive.

PS. The title "Tailoring supercurrent confinement in graphene bilayer weak links" leads me to believe that multiple samples were measured. If so, could the authors clearly indicate which figures correspond to the additional samples?

Reply to reviewers

Reviewer #1:

“The authors investigated the supercurrent in bilayer graphene weak links using a QPC-like geometry. They used the split-top gates to create one-dimension constrictions. The ability to control the dimension of the weak links from two to one dimension allows them to tune critical currents as well as density profile of the supercurrent. The authors also studied the interference pattern of the supercurrent vs magnetic field as the weak link dimension changes from two to one dimension by varying split gate voltage and back gate voltage. The results agree well with their theoretical model.”

- 1- “The ability to create 1D constriction in this experiment relies on an insulating state in BLG which can be induced by displacement field. However, such insulating state is very hard to achieve in BLG due to small band gap generated by displacement field and potential fluctuations from disorders. The authors should estimate the size of the band gap from displacement field (see Nature 459, 820-823 (2009)) and the potential fluctuation from a residue charge inhomogeneity and see if the band gap is larger than the potential fluctuation or not.”

We thank the reviewer for pointing out the importance of the comparison between the potential fluctuation and the band gap size. Indeed, we see that the expected values are in complete agreement with our experiment. We provide quantitative estimation of the gap induced by the displacement field that breaks the lattice symmetry underneath the split-gate using the formula provided in ref [31-32] (from the main text) and obtain a band gap of $E_g \sim 85.4$ meV at back and split-gate voltage values of 8 V and -7.6 V respectively which corresponds to a displacement field of $D \sim 0.56$ V/nm following Zhang *et al.* (ref. [36] from the main text). If we compare the band gap value to the potential fluctuation from our residual charge inhomogeneity $n_{res} \sim 2.6 \times 10^{10}$ cm⁻², we find that this energy value corresponds to an excitation of ~ 1 meV. This energy value is much smaller than the estimated band gap. Therefore, we have decided to integrate a short discussion in the supplementary information (page 4-5 and ref [8,9] have been added in the supplementary information too):

” We provide quantitative estimation of the gap induced by the displacement field that breaks the lattice symmetry underneath the split-gate using the formula provided in [8,9]. The gap should be larger than the potential fluctuation coming from the residual charge carrier inhomogeneity $n_{res} \sim 2.6 \times 10^{10}$ cm⁻² which corresponds to an excitation of $E \sim 1$ meV (a band gap of 1 meV can be obtained by applying back and top gate values of for example $V_{BG} = 0.14$ V and $V_{SG} = -0.13$ V respectively). At back and split gate values of $V_{BG} = 8$ V and $V_{SG} = -7.6$ V respectively, which correspond to a displacement field of $D \sim 0.56$ V/nm following Zhang *et*

al. [15], we obtain an energy band gap of $E_g \sim 85.4$ meV, *i.e.* a value much larger than the potential fluctuation.”

- 2- “From Fig. 3a, b in the supplementary information, Fabry-Perot oscillations exists in NP_nN regime and the period of these oscillations corresponds to the cavity underneath the split gates. Doesn’t this mean that BLG under the split gates in the NP_nN regime is not completely insulating since electrons can still propagate through it. Therefore, the device does not exhibit 1D constriction in the NP_nN regime. “

Fig. 3 in the supplementary information displays the differentiated gate map $dG/dV_{BG}(V_{BG}, V_{SG})$ in the normal state which highlights fine Fabry-Pérot oscillations with periodicities corresponding to the cavities in which the carriers interfere. In the NP_pN region interferences occur mainly in the cavities formed by the split-gate. The interferences are visible as parallel lines in the gate map which are tuned by both back- and top gate (shown in red in Fig. 3a).

It is true that the NP_nN region does show features which look like Fabry-Pérot interferences as well. While the almost horizontal parallel lines can be identified as oscillations between the contacts (cavity size L_{cav} of about 1 μm), they are nearly independent of V_{SG} , indicating trajectories through the channel of the QPC which are only weakly tuned by the stray fields of the split-gate. If these interferences would take place by transport of charge carriers through the top-gated cavity, the interferences should be tuned in a stronger way by V_{SG} as one can see in the PPP (and NP_pN) region.

Furthermore, we can identify non-parallel larger stripes in the NP_nN region. When we extract the cavity size by assuming Fabry-Pérot interferences, we find $L_{cav} < 150$ nm at a back-gate value of 8 V which is much less than the size of the cavity corresponding to the split-gate width $L_{SG} \approx 300$ nm.

So we conclude that these features do not correspond to the cavity created by the split-gate. The sudden drop of the supercurrent together with the magneto-interference pattern transition combined with our analytical and numerical models confirms that the supercurrent is confined by the electrostatically defined constriction and that no current passes underneath the split-gate.

- 3- “The authors attribute the step-wise decrease of the critical current to ballistic supercurrents through QPC (Fig 3e in main text). However, they do not observe any signature of 1D subbands in the normal state. Do you have any explanation why 1D subbands would be more pronounced in the superconducting state? Can the “step-wise decrease” be the consequence of Fabry-Perot oscillations (It just happened to look like steps when in fact it’s due to oscillations of resistance in the normal state). Figure 3d in the supplementary information also shows oscillations in critical current with similar periods as the one in Fig 3e of the main text.”

Here reviewer #1 (and to some extent reviewer #3) points out the fact that the step-like features observed in the NP_nN area may appear due to the presence of Fabry-Pérot interferences. As aforementioned, these oscillations of the differentiated conductance which could be seen as Fabry-Pérot interferences do not correspond to any possible formed cavity within our device. As pointed out by referee #1, each of these features seen in the critical current correspond to one in the normal state, unlike what we mentioned in the main text. Therefore, we have decided to replace in the main text the following sentence:

“Importantly, despite the absence of signs of 1D subband formation while shrinking the constriction in the normal state, the critical current decreases in a step-wise fashion (see Fig. 3e) as predicted for ballistic supercurrents in quantum point contacts [38–40]. “

by:

“We note that, on the presented device, we observe features visible in the normal and the superconducting state which could be related to quantized conductance and supercurrent (see supplementary information) as predicted for ballistic supercurrents in quantum point contacts [38–40].”

We have measured these features in our smaller device (with a constriction defined by the split-gate of $w \sim 60\text{nm}$), but not in our larger device (with a constriction defined by the split-gate of $w \sim 150\text{nm}$). It sounded important to us to mention these features in the main text but since the data are not so clear, a conclusive explanation remains beyond the scope of our paper. Since it is not the real focus of our work, we have decided to remove Fig. 3e and place it together with normal and superconducting curves in the supplementary information (now Fig. 6) as well as a series of similar plots under various back gate conditions (in Fig. 7) in a new subsection entitled “C. Any signs of quantized supercurrent?” from part “iV. Additional superconductivity data” and reads:

“In our experiments, we clearly observe Fabry-Pérot interferences highlighting ballistic transport of the charge carriers all across the length of the device. Therefore, one might expect to observe quantized conductance while inducing a constriction in a two-dimensional system [21,21]. This phenomenon has been extensively studied in particular in AlGaAs/GaAs heterostructures [23]. Until now, step-like features in the conductance have been observed in graphene (both single and bilayers) but none of them quantized in the expected value of $4e^2/h$ [24-33] (the prefactor 4 refers to the spin and valley degeneracy). For our configuration using only one overall back-gate and a local split-gate, both the tuning of the Fermi energy and the opening of the gap cannot be fully independently controlled. At V_{BG} and V_{SG} sufficiently large to form the constriction and confine the supercurrent, the charge carrier density within the constriction which is mostly influenced by the back-gate compared to the stray field generated by the split-gate, might appear to large and the confinement not strong enough to clearly form 1D subbands and therefore both quantized conductance and supercurrent appear to be hard to observe. Therefore the picture drawn for more conventional semiconductors might not be applicable [34-36]. In our case we observe features in the normal and superconducting state conductance although non-quantized, here at $V_{\text{BG}} = 8.6\text{ V}$ (see Fig. 6a and b). At this back-gate voltage value and $V_{\text{SG}} \sim -8.5\text{ V}$ the stray field generated by the split-gate starts to balance the action of the back-gate on the charge carrier density. As we can see, the minimum conductance reaches $24e^2/h$ corresponding to 6, four times degenerated, opened channels and a resistance of $\sim 1\text{ k}\Omega$. We note that these features are observable in a large back-gate range (see Fig. 7). What we define as signs of quantized supercurrent (together with signs of quantized conductance) was measured in our shorter devices with $w \sim 65\text{ nm}$ split-gate distance but not in our longer device with a wider constriction ($w \sim 150\text{ nm}$ split-gate distance, not shown here) which showed all properties of supercurrent confinement (in both amplitude and magneto-interferometric pattern) that we present in this work.”

A large set of new references has also been added.

New Fig. 6a and b (plus caption) of the supplementary info:

FIG.6. Step-like features in the supercurrent and in the normal state conductance. **a** Differential resistance $dV/dI(V_{SG}, I)$ map as a function of split-gate voltage V_{SG} and current I zoomed-in on the NP_nN region, revealing a step-wise reduction of the critical current I_c (red curve). **b** Critical current I_c (red) and normal state conductance G (blue) vs V_{SG} at $V_{BG} = 8.6V$.

New Fig. 7 (plus caption) of the supplementary info:

Fig.7. Step-like features in the supercurrent under various back-gate conditions. Critical current I_c and conductance G (top) and differential resistance $dV/dI(V_{SG}, I)$ maps (bottom)

showing step-like features in the supercurrent for various back-gate voltages, from left to right $V_{BG}=8.6, 9, 9.1, 9.2$ and $9.3V$ respectively.

- 4- “The authors mention Fig 4c at the last line of page 6 of the main text but there is no Fig 4c.”
We have corrected this typo and changed Fig.4c to Fig.4b.

Reviewer #2:

“The manuscript by Kraft et al. reports a systematic and detailed investigation of the spatial and amplitude variation of the superconducting critical current (I_c) in hBN-encapsulated bilayer graphene obtained via the application of back- and top-gate voltages. The results obtained are technically correct and the analysis of the measurement data is indeed very detailed. However, there have been many studies recently which have explored the possibility of controlling the superconducting I_c in graphene-based Josephson devices by tuning the charge carrier type and density through the application of gate voltages. Other research groups have demonstrated changes in the graphene normal-state conductance from a few-mode to a single-mode channel (e.g. P. Rickhaus et al., *Nano Lett.* 15 (9), 5819–5825 (2015)), as well as investigated the spatial variation of I_c with bias voltage in superconducting S/N/S junctions of different lengths where hBN-encapsulated graphene is the weak link (e.g. I.V. Borzenets et al., *Phys. Rev. Lett.* 117, 237002 (2016) or M.B. Shalom et al., *Nat. Phys.* 12, 318-322 (2016)).

From this point of view, the manuscript by R. Kraft et al. seems a natural extension of similar studies already reported in the literature for single-layer graphene to the case of bilayer graphene. Although it can be argued that this work is different compared to other studies on the modulation in I_c as a function of carrier density and type in bilayer graphene (e.g. M.T. Allen et al., *Nat. Phys.* 12, 128-133 (2016)) and that a systematic study of the spatial variation of I_c is addressed, I think that it does not meet the elements of novelty and originality necessary for publication in *Nature Communications* and therefore I recommend its publication in a more specialist journal.”

We disagree with reviewer #2 on the critics regarding the novelty of our work presented in our manuscript. The reviewer first cites P. Rickhaus *et al.*, *Nano Lett.* 15 (9), 5819–5825 (2015) which does not involve any spatial control of superconductivity and confinement based on gap engineering in graphene bilayer, then refers to the work of I.V. Borzenets *et al.*, *Phys. Rev. Lett.* 117, 237002 (2016), i.e. ref [29], and M.B. Shalom *et al.*, *Nat. Phys.* 12, 318-322 (2016), i.e. ref [26] which does not involve any confinement of supercurrent and finally mentions the work of M.T. Allen *et al.*, *Nat. Phys.* 12, 128-133 (2016), i.e. ref [27] (please note that one of the co-authors of our manuscript has been part of this work) where edge supercurrent flow is detected near the charge neutrality point in both single and bilayer graphene using Dynes and Fulton’s technique based on magneto-interference (R. Dynes & T.A. Fulton, *Phys. Rev. B* 3, 3015-3023 (1971), i.e. ref. [42] in our manuscript). In these works, the supercurrent is tuned solely by an overall back-gate. None of the references cited by reviewer #2 do involve local gate control of supercurrent in its amplitude and spatial distribution. Here we have used a quantum point contact geometry to show that supercurrent could be confined which has not been studied anywhere else to our knowledge. In addition, we demonstrate that gapped bilayer graphene may not be disturbed by any edge current.

“Lastly, I think the authors may find useful to address the following minor points which would make the text of the manuscript clearer to the reader:

- 1- In paragraph on page 2 starting with “As a consequence, the bipolar region..” the text would be clearer if the authors explained all the labels used (NPnN, NPpN etc.) – which are actually clearly defined only in the caption of Figure 2.”

We follow the request of reviewer #2 and insert the text which was in the caption of Fig. 2 in the paragraph which describes the different doping regions in the QPC geometry. Now the first sentence of the last paragraph of page 2 reads: “The schematics in Fig.2d summarize the different scenarios which govern the behaviour of such an electrostatically induced constriction, *i.e.* the formed 1D constriction area NP_nN, the unipolar regime NNN and the non-uniform NP_pN junction.”

- 2- “The scale bar in Figure 2c does not look correct, as it seems that Figure 2c should contain some areas coloured in green in Figure 2b – which should correspond to a resistance value of 0.3-0.4 k Ω according to the scale bar in Figure 2b.”

Since this figure is mainly utilized to emphasize the resistive part and not used to show any quantitative value, we have not displayed the map “full scale”. Now we have corrected this and we show the figure “full scale”, *i.e.* we have changed the scale of Fig. 2c which was from 0 to 200 Ω , now to 0 to 500 Ω .

New Fig. 2c (No change in the caption):

- 3- “In the caption of Figure 3e, I think the authors mean ‘differential resistance’ other than just ‘resistance’.”

Reviewer #2 is correct. We have changed the caption text with “differential resistance” instead of “resistance”.

- 4- “In the paragraph starting with “First, a beating pattern appears..” on page 5 the authors should clarify that the evolution from a Fraunhofer-like pattern into a bell-shaped I_c Vs. H curve happens for increasing gate voltages, as this only becomes clear to the reader after looking the captions of Figure 4a.”

To clarify our explanation, we have revised the sentence which describes the evolution of the magneto-interference pattern as follows: “By increasing V_{SG} , *i.e.* the confinement, a

progressive change of the interference pattern is observed as the split-gate is tuned and the 1D constriction forms. First, a beating pattern appears, resembling Fraunhofer-like interference (upper panel) when the system remains two-dimensional.”

- 5- “In the caption of Figure 4, the split-gate voltage values should be negative, in order to be consistent with what reported in Figures 4a and 4b as well as in Figure 1c.”

We have changed the split-gate voltage values to negative.

- 6- “At the end of page 6, the authors explain how $I_{c\text{norm}}$ is calculated, but they never define $I_{c\text{norm}}$ explicitly. This should be done, as $I_{c\text{norm}}$ is used for the first time in the caption of Figure 5 but without any definition.”

In our original manuscript we define I_c^{norm} as such: “In Fig. 4b we can observe a map of the critical current I_c (left panel) as well as the critical current normalized with the maximum critical current (at $B = 0$) I_c^{norm} (right panel) as a function of magnetic field B and split-gate voltage V_{SG}, \dots ”. We now define I_c^{norm} as follows “In Fig. 4b we can observe a map of the critical current I_c (left panel) as well as the critical current normalized to the maximum critical current (at $B=0$) $I_c^{\text{norm}} = I_c/I_c(0)$ (right panel) as a function of magnetic field B and split-gate voltage V_{SG}, \dots ”.

- 7- “In the last section of the supplementary information, the authors state that the temperature used in the numerical simulation is higher than the actual measurement temperature, as the relevant scale is the thermal length. Using a lower temperature in the numerical code, however, should have an effect at least on the amplitude of the calculated I_c and perhaps on its spatial dependence as well. It would help if the authors could further clarify on this point and perhaps add to the SI file another $I_{c\text{norm}}$ versus applied field contour plot, similar to that shown in Figure 5c, but calculated at a different temperature (e.g. $k_B T = \Delta_0/10$).”

We have repeated the simulations, this time at $k_B T = \Delta_0/20$. No strong variations have been seen in particular on the interference pattern transition corresponding to a change in spatial distribution of the supercurrent. At lower temperature, small features are slightly more pronounced. We have changed Fig.5c in the main text as well Fig.10 of the supplementary information and change the value of $k_B T = \Delta_0/20$ in the text.

New Fig. 5c (No change in the caption):

New Fig.10 of the supplementary info (No change in the caption):

Reviewer #3:

The authors study encapsulated bilayer graphene contacted by superconducting titanium/aluminium electrodes. They fabricate a split gate on top of the sample (Figure 1), which eventually allows them to study superconducting current through a 1D constriction. They measure the normal state resistance and the critical current vs the back gate and the split gate in Figure 2. Different regimes are identified, depending on whether the regions of the channel and that under the split gate are P or N doped. In particular, the maps of Figure 2 allow them to find the range of parameters in which the region under the gate is gapped and depleted, but the channel is doped and conductive. This situation occurs when the back gate voltage is high enough (beyond about 6 V), and the split gate voltage is negative enough to shut off the conductance bypassing the channel formed by the split gate. In this regime, the residual critical current of the order of 10-100 nA is observed (Figure 3d,e).

At this point the author make a rather ambitious claim that the critical current is quantized: “Importantly, despite the absence of signs of 1D subband formation while shrinking the constriction in the normal state, the critical current decreases in a step-wise fashion (see Fig. 3e) as predicted for ballistic supercurrents in quantum point contacts [38–40].” I totally agree with them that without observing the quantized conductance, it is strange to expect the critical current quantization. To substantiate their claim, I believe that it is crucial to demonstrate this behavior in more than one sample. Also, it would be important to show explicitly how the map of Figure 3e is converted to a graph of critical current vs gate voltage. Finally, the normal state resistance in the range shown in Figure 3e appears to be about 500 Ohm, corresponding to conductance of about $50 e^2/h$. It seems inconceivable that the critical current would be quantized and show only a few first few quantized steps as it may be suggested by the shape highlighted in Figure 3e.

The rest of the paper (Figure 4) and most of the supplementary information describe the magnetic interference pattern. The similarity between the data and numerical simulations appears very conclusive.

PS. The title “Tailoring supercurrent confinement in graphene bilayer weak links” leads me to believe that multiple samples were measured. If so, could the authors clearly indicate which figures correspond to the additional samples?

Reviewer #3 appears to have similar remarks on the quantized supercurrent as reviewer #1. In order to address the reviewer's concerns we the added part on the signs of quantized supercurrent in the supplementary info. We note that reviewer #3 thinks that Fig. 3e, which was in the original draft and now appears in the supplementary information, reaches the normal state at the high bias values. However, since the data here is shown as a function of current and not voltage, the biasing is not large enough to reach the normal state resistance (in terms of energy the biasing is still within twice the superconducting gap, *i.e.* below $\sim 200 \mu\text{eV}$, in the scale of the current displayed in this color map). So the normal state differential resistance above twice the superconducting gap corresponds to the normal state resistance when applying a small magnetic field ($\sim 20 \text{ mT}$) to fully break the superconductivity. Therefore, the normal state conductance, as now shown if Fig. 6b of the supplementary information, drops to $24e^2/h$. Finally, as we mention in the new part on the sign of quantized supercurrent, we have seen these features in our shorter device with $w \sim 65 \text{ nm}$ split-gate distance. Our longer device with a wider constriction $w \sim 150 \text{ nm}$ split-gate distance (not shown here) showed all properties of confinement that we present in this work (we also have recently measured devices with other confinement configurations which fully confirms our findings presented in this paper), but the signs of quantization.

Additionally:

- We have added under brackets the thickness of the two (bottom and top) hBN multilayers in the caption of Fig.1. Now it reads:
“...(with a bottom and top hBN multilayer of ~ 35 and $\sim 38 \text{ nm}$ thick) on a pre-patterned overall back-gate (BG) covered with a 20 nm thick Al_2O_3 and a split-gate (SG) on top of the heterostructure. The superconducting leads are edge connected to the mesa. The width $W =$

3.2 μm and length $L = 950 \text{ nm}$ while the distance between the two fingers of the split-gate $w \sim 65 \text{ nm}$ (and $d' \sim 38 \text{ nm}$ and $d'' \sim 55 \text{ nm}$). **c** AFM image of the device. Scale bar is $1 \mu\text{m}$.”

- We have found that our extracted contact resistance was not properly calculated. We have changed the text of the part “II. Normal state characterization” as follows: “Fig.1a shows both normal state resistance R and conductance G as a function of back-gate voltage V_{BG} and charge carrier density n , while Fig.1b displays the electron conductivity (minus the estimated contact resistance calculated below) vs charge carrier density. The contact resistance per contact is estimated as $R_C = (R - R_Q)/2$, where the quantum resistance R_Q is subtracted from the measured resistance R . The quantum resistance $R_Q = (h/ge^2)(1/M)$ is defined as the resistance set by the ballistic limit of all contributing conductance modes $M = W/\lambda_F/2$, where $\lambda_F = 2\pi/k_F = 2\pi/(\pi n)^{1/2}$ is the Fermi wavelength at charge carrier density n and $g = 4$ accounts for the spin and valley degeneracy. At high charge carrier density $n \sim 4 \times 10^{12} \text{ cm}^{-2}$ ($M = 361$ and $R_Q = 18\Omega$) a resistance of $R = 90\Omega$ is measured, yielding the contact resistance $R_C = 36\Omega$ and contact resistivity $\rho_C = R_C W = 115\Omega\mu\text{m}$, comparable to the values given by Wang et al. [1].” As a consequence, the Fig. 1b has been replotted and corrected. The extracted value for the residual carriers is now $2.6 \times 10^{10} \text{ cm}^{-2}$.

New Fig.1b of the supplementary information:

b

- Change in Fig.3 of the supplementary information: numbers on the gray-scale map bar were wrong and have been corrected.
- We have refined the analytical model that was based in the original version of the manuscript on the approach from Barzykin, V. & Zagoskin, A.N. *Coherent transport and nonlocality in mesoscopic SNS junctions: anomalous magnetic interference patterns*. Superlatt. Microstruc. 25, 797 (1999) (ref. [37] of the supplementary information), see subsection entitled “A. Analytical model: Long junction” from the part V entitled “Analytical and numerical model” of the supplementary information. In the new version, we have recalculated the critical current in the long-junction limit by going beyond the approximation of that reference. We have also corrected the phase difference in Eq. (9) of supplementary information. The modifications concern the geometrical weight of the Andreev trajectories (cf. for example Eq. (2) in the previous and new versions) and the overall scale of the

interference pattern as a function of magnetic field (the latter was anyway used as a fitting parameter). The fit using our refined formula is presented in new Fig. 5b:

New figure 5b

Old figure 5b

Importantly, the modifications in our analytical expressions do not produce any qualitative change of the behavior of the interference pattern for the QPC setup. Moreover, as one can see from the comparison of the new version of Fig. 5b with the previous version, the two fits are almost indistinguishable. We have also re-written some parts of the corresponding subsection of supplementary information to improve the presentation.

Reviewers' comments:

Reviewer #1 (Remarks to the Author):

I find the changes that the authors made satisfy the concerns I raise in my review. I now believe that this work can be published in Nature Communications.

I have some minor comments the authors should address prior to publication:

- 1) In Figure 2 in the main text, the authors should give a value of current bias used in the resistance map measurement (fig 2a, b, and c).
- 2) In Fig. 5c, the definition of t of the label on the y-axis ψ_{SG}/t is not given in the main text.
- 3) Is it possible to relate ψ_{SG}/t to the measured V_{SG} in order to check if the transition from beating to non-beating patterns agrees with the theory?
- 4) In the supplementary Information on page 2, one of the sentences reads "In a subsequent step, split-gates are fabricated in the fashion". Do you mean "in the similar fashion"?

Reviewer #2 (Remarks to the Author):

The authors have made a significant effort in addressing all the questions and remarks of the referees, and inserted additional data which make the manuscript and its discussion more solid and sound from a scientific point of view.

However, it must be noted, that reviewer #3 pointed out the need for showing evidence for a quantised supercurrent in more than a single device, which I agree it is a very important point. In response to this remark, the authors simply clarify that features of quantised conductance were only observed for a single device with a split-gate distance of ~ 65 nm (they state "as we mention in the new part on the sign of quantized supercurrent, we have seen these features in our shorter device with $w \sim 65$ nm split-gate distance" in the rebuttal letter), therefore leaving the issue of reproducibility in more than a single device raised by reviewer #3 still open. Apart from this question that has to be still addressed, I think the manuscript can now be considered more suitable for publication in Nature Communication.

Reviewer #3 (Remarks to the Author):

I have read the authors' replies to all referees. I am glad that the incorrect claim of a quantized critical current (criticized by refs. 1 & 3) has been mostly moved to the supplementary. I am surprised that they kept the following remnant of that claim: "We note that, on the presented device, we observe features visible in the normal and the superconducting state which could be related to quantized conductance and supercurrent (see supplementary information) as predicted for ballistic supercurrents in quantum point contacts [38–40]." On that level, it is up to the authors to decide whether they want to cling to that statement. At least most of it will not be presented in the text of the paper. It would be lovely if they explained why their sample appears to show no edge conductance, while similar samples from the Manchester group (ref. [50]) do show pronounced edge supercurrent.

Reply to reviewers

Dear Editor,

We wish to thank the reviewers once more for their constructive remarks and comments. Here, we address each of the reviewer's questions and issues point by point (modification-additional text in red).

Reviewer #1 (Remarks to the Author):

I find the changes that the authors made satisfy the concerns I raise in my review. I now believe that this work can be published in Nature Communications. I have some minor comments the authors should address prior to publication:

1) In Figure 2 in the main text, the authors should give a value of current bias used in the resistance map measurement (fig 2a, b, and c).

As we have mentioned in the method section, we have used ac excitation for the lock-in detection between 1 and 10 μV . For Fig 2a (normal state data), within this resistance map the largest voltage drop at the sample was 2.5 μV while the maximum current was 2.6 nA. For Fig. 2b (and c) (*i.e.* superconducting state) the voltage drop at the sample was at a maximum of 2 μV while the maximum current was 2.5 nA. We added the detailed values of the maximum measured ac current in the caption of figure 2.

2) In Fig. 5c, the definition of t of the label on the y-axis ψ_{SG}/t is not given in the main text.

φ_{SG}/t represents the strengths of the on-site potentials introduced on the split-gate in units of the intralayer hopping constant t . We have added this information in the text (in the last sentence before the conclusion section) as follows:

"We finally show tight-binding simulations using Kwant package [53] of I_c as a function of magnetic field B and split-gate strength φ_{SG} (in units of the intralayer hopping constant t) in Fig. 5c (see supplementary information for details) which are in good qualitative agreement with our experimental data of Fig. 4b."

3) Is it possible to relate ψ_{SG}/t to the measured V_{SG} in order to check if the transition from beating to non-beating patterns agrees with the theory?

Unfortunately realistic and quantitative modeling of electrostatic potentials caused by a constriction in similar geometries is beyond the current state-of-the-art tight binding simulations. We therefore limit our consideration to a qualitative comparison.

4) In the supplementary Information on page 2, one of the sentences reads “In a subsequent step, split-gates are fabricated in the fashion”. Do you mean “in the similar fashion”?

We thank the reviewer for pointing this out. We have corrected it with “in the similar fashion” in the text.

Reviewer #2 (Remarks to the Author):

The authors have made a significant effort in addressing all the questions and remarks of the referees, and inserted additional data which make the manuscript and its discussion more solid and sound from a scientific point of view. However, it must be noted, that reviewer #3 pointed out the need for showing evidence for a quantised supercurrent in more than a single device, which I agree it is a very important point. In response to this remark, the authors simply clarify that features of quantised conductance were only observed for a single device with a split-gate distance of ~ 65 nm (they state “as we mention in the new part on the sign of quantized supercurrent, we have seen these features in our shorter device with $w \sim 65$ nm split-gate distance” in the rebuttal letter), therefore leaving the issue of reproducibility in more than a single device raised by reviewer #3 still open. Apart from this question that has to be still addressed, I think the manuscript can now be considered more suitable for publication in Nature Communication.

We thank Reviewer #2 for these positive comments. As we have mentioned in the text, we have measured two devices with similar geometry (QPC, one shorter and with small distance between the split-gates, shown here and a longer device) and both showed the similar behavior while confining the supercurrent. Only the signs of quantized supercurrent were seen in the smaller device presented in this article as we mention in our reply to reviewers and described in section **IV.C**. We have also observed the effect of confinement in another geometry (with side-top-gates, i.e. forming a 1D channel the entire length of the device) which ends to a very similar magneto-interferometric pattern. We will describe these experiments in another paper.

Reviewer #3 (Remarks to the Author):

I have read the authors’ replies to all referees. I am glad that the incorrect claim of a quantized critical current (criticized by refs. 1 & 3) has been mostly moved to the supplementary. I am surprised that they kept the following remnant of that claim: “We note that, on the presented device, we observe features visible in the normal and the superconducting state which could be related to quantized conductance and supercurrent (see supplementary information) as predicted for ballistic supercurrents in quantum point contacts [38–40].” On that level, it is up to the authors to decide whether they want to cling to that statement. At least most of it will not be presented in the text of the paper. It would be lovely if they explained why their sample appears to show no edge conductance, while similar samples from the Manchester group (ref. [50]) do show pronounced edge supercurrent.

We thank reviewer #3 to raise the issue of edge currents. We address it in a new section (section **VI** entitled “Effect of the edge currents on the magneto-interferometric pattern” and added 8 references) in the supplementary information. First by implementing our analytical model with edge channels (figure 12 presenting the geometry of the system), we introduce a correction factor which

accounts for the edge channel. Figure 13a shows the correction factor itself versus the magnetic field for different values of the transmission ratio T_e/T_q . At large T_e/T_q the correction factor shows the expected SQUID magneto-interferometric pattern. When we apply the correction factor to the QPC, *i.e.* adding edge channels, the originally monotonic decay of I_c while B increases appears to be modulated (Figure 13b, already at a T_e of about 1% of T_q). This shows that the presence edge currents should strongly affect the magneto-interferometric pattern. In our case this would mean that we should be sensitive to at least 200 pA edge current amplitude (with $I_c \sim 20$ nA).

Second, our measurements were performed at much higher displacement field than in the experiment reported in Ref [50] in the main text (one order of magnitude higher). This is the major reason why we do not observe any sign of edge currents. In our experiments, the edge states do not contribute to the magneto-interference pattern and this is fully consistent with the theoretical expectation of Ref [44] in the supplementary information. Edge states appear in bilayer graphene only for certain orientation of the edges and are not topologically protected against disorder and edge imperfections. This should be contrasted with the topologically protected helical edge states in two-dimensional topological insulators for example. As a result, even for the most favourable configuration for the edge currents, *i.e.* zigzag orientation, any disorder eventually leads to localisation of edge states. The localisation length depends on the degree of disorder and on the band gap value in the bulk (see Fig. 3 of Ref. [44]). The absence of edge contribution to the total current could be explained by the edge-state localisation in the case of large band gap as reported in [44]. The characteristic value of the localisation length was found to be about tens of nanometers, which is much smaller than the top-gated region in our experiment, thus implying strong suppression of edge currents.

Please find the new added text plus figures in section **VI** of the supplementary information.

REVIEWERS' COMMENTS:

Reviewer #2 (Remarks to the Author):

I think the authors have properly addressed all the questions and points made by the referees in the previous reviewing stages, and their manuscript is currently suitable for publication in Nature Communications.